# Experimental demonstration of optical stochastic cooling

J. Jarvis[1✉], V. Lebedev[1✉], A. Romanov[1], D. Broemmelsiek[1], K. Carlson[1], S. Chattopadhyay[1,2,3], A. Dick[2], D. Edstrom[1], I. Lobach[4], S. Nagaitsev[1,4], H. Piekarz[1], P. Piot[2,5], J. Ruan[1], J. Santucci[1], G. Stancari[1] & A. Valishev[1]

Particle accelerators and storage rings have been transformative instruments of discovery, and, for many applications, innovations in particle-beam cooling have been a principal driver of that success[1]. Stochastic cooling (SC), one of the most important conceptual and technological advances in this area[2–6], cools a beam through granular sampling and correction of its phase-space structure, thus bearing resemblance to a 'Maxwell's demon'. The extension of SC from the microwave regime up to optical frequencies and bandwidths has long been pursued, as it could increase the achievable cooling rates by three to four orders of magnitude and provide a powerful tool for future accelerators. First proposed nearly 30 years ago, optical stochastic cooling (OSC) replaces the conventional microwave elements of SC with optical-frequency analogues and is, in principle, compatible with any species of charged-particle beam[7,8]. Here we describe a demonstration of OSC in a proof-of-principle experiment at the Fermi National Accelerator Laboratory's Integrable Optics Test Accelerator[9,10]. The experiment used 100-MeV electrons and a non-amplified configuration of OSC with a radiation wavelength of 950 nm, and achieved strong, simultaneous cooling of the beam in all degrees of freedom. This realization of SC at optical frequencies serves as a foundation for more advanced experiments with high-gain optical amplification, and advances opportunities for future operational OSC systems with potential benefit to a broad user community in the accelerator-based sciences.

Particle accelerators are invaluable scientific tools that have enabled a century of advances in high-energy physics, nuclear physics, materials science, fusion, medicine and beyond[1]. In many applications, high-brightness particle beams are required, and for those relying on storage rings (for example, particle colliders, light sources, and light-ion and heavy-ion rings), beam cooling is an indispensable element of the accelerator's design and operation. Beam cooling constitutes a reduction of the six-dimensional phase-space volume occupied by the beam particles or, equivalently, a reduction in the thermal motion within the beam. In the case of colliders, cooling increases luminosity through the reduction of beam emittances and is essential for combatting intrabeam scattering (IBS) and other diffusion mechanisms[11,12]. Cooling also enables and supports a broad range of other applications in atomic, particle and nuclear physics, including the efficient production of antihydrogen for tests of charge, parity, time-reversal (CPT) symmetry and gravity[13–15], internal-target experiments for precision measurements of resonance masses and widths[16], and the production and cooling of both stable and radioactive ion species for precision measurements of states and interactions[17,18].

There is a wide array of application-specific cooling techniques[19,20]. One of the most common is synchrotron radiation (SR) damping, which results from the beam's emission of SR in bending magnets and the subsequent replenishment of this energy loss by radio-frequency accelerator cavities[21]. For electron–positron colliders, as well as proposed hadron colliders on the energy frontier (for example, the Future Circular Collider), adequate cooling is already present owing to SR damping[22,23]; however, for hadrons at energies below about 4 TeV, SR damping times at the collision energy are too long for practical use, and effective cooling requires an engineered system.

For such systems, two primary families of cooling methods can be considered: electron cooling (EC) and stochastic cooling (SC)[2,3,24–26]. In EC, a hadron beam's temperature is reduced as the particles thermalize through Coulomb scattering with a velocity-matched, low-temperature electron beam. Unfortunately, the scaling of EC with beam energy becomes especially unfavourable for relativistic beams. EC may be feasible for the planned Electron Ion Collider (EIC) at Brookhaven National Laboratory, which has an anticipated operational ceiling of 275 GeV (protons), but the potential for EC systems beyond this energy is uncertain[27,28].

SC, first suggested by S. van der Meer in 1968, was a key technology in the success of proton–antiproton colliders. It was instrumental in the discovery of the W and Z bosons in 1983, as it enabled the accumulation of a sufficient number of antiprotons with the required beam quality, and a year later, van der Meer received a share of the Nobel

[1]Fermi National Accelerator Laboratory, Batavia, IL, USA. [2]Department of Physics, Northern Illinois University, DeKalb, IL, USA. [3]SLAC National Accelerator Laboratory, Menlo Park, CA, USA. [4]Department of Physics, The University of Chicago, Chicago, IL, USA. [5]Argonne National Laboratory, Argonne, IL, USA. ✉e-mail: jjarvis@fnal.gov; val@fnal.gov

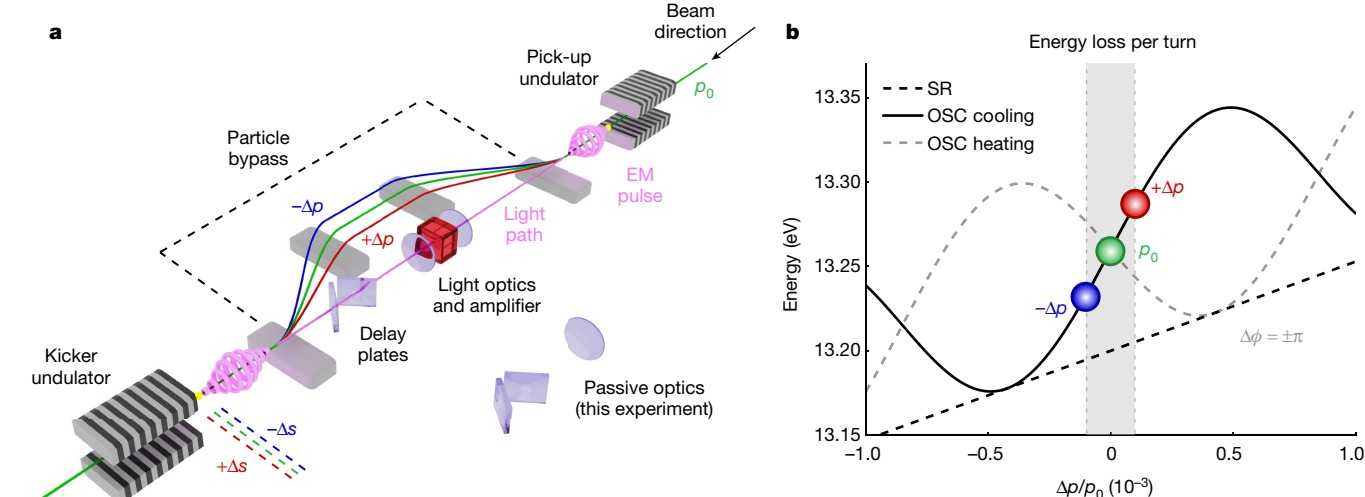

**Fig. 1 | Conceptual schematic and operating principles of a transit-time OSC system. a**, Each particle produces a pulse of electromagnetic radiation as it transits the pick-up undulator. A magnetic bypass separates the beam and light and encodes each particle's phase-space error on its arrival delay at the kicker undulator. Trajectories for particles with positive (red) and negative (blue) momentum deviations are shown, along with their corresponding arrival delays, relative to the reference particle (green). The pick-up radiation is amplified (or not) and focused into the kicker undulator. The particles'

interaction with the pick-up radiation inside the kicker produces corrective energy kicks when the system is tuned for the cooling mode. **b**, Example of energy loss per turn with (black solid line) and without (black dashed line) OSC as a function of a particle's relative momentum deviation $\Delta p/p_0$. Detuning the delay system by half of the fundamental wavelength (a shift of $\Delta\phi = \pm\pi$ in the radiation phase) places the OSC system in a heating mode (grey dashed line). The grey shaded region corresponds to the relative r.m.s. energy spread for the reported configuration (about $10^{-4}$).

Prize in Physics for SC and its role in the discovery[2–5]. Since then, SC has been used to broaden the science reach of many facilities, including the Tevatron collider, where it enabled the discovery of the top quark, the Relativistic Heavy Ion Collider, which is the first collider to use SC operationally at the collision energy, and the Experimental Storage Ring[6,11,12,18,29].

In a conventional SC system, statistical fluctuations in the beam are sampled using electromagnetic pick-ups (antennas) operating in the microwave regime with bandwidths up to about 10 GHz. The resulting signals are then amplified and applied to electromagnetic kickers in a negative-feedback system, resulting in cooling of the circulating beam. As SC is based on the sampling of random fluctuations, the bandwidth of the integrated system and the particle density in the beam determine the number of passes required for cooling, and thus the achievable cooling rate[25,26]. The extension of SC to optical frequencies with a subsequent increase of bandwidths (~$10^{13}$ Hz) could increase the achievable cooling rates by three to four orders of magnitude and, for example, enable direct cooling of high-density proton and antiproton bunches between 0.25 TeV and 4 TeV. At present, two possible methods have been proposed for such an implementation: optical stochastic cooling (OSC) and coherent electron cooling (CEC)[7,8,30].

CEC is being developed as a candidate for beam cooling of hadrons in the planned EIC[29]. It uses an electron beam as the pick-up, kicker and amplifier[30,31,32,33]. In contrast, OSC uses free-space electromagnetic waves as the signalling medium, magnetic undulators to couple the radiation to the circulating particle beam and optical amplifiers for signal amplification. In both cases, the underlying physics of the SC method remains unchanged in the transition to optical frequencies. OSC was first suggested nearly three decades ago, and although several proposals for its implementation were made, both for operational hadron colliders and low-energy electron rings, the concept has not been validated experimentally up to now[9,34–38]. Here we describe the experimental realization of OSC. This result constitutes a successful demonstration of beam cooling with a SC technique at optical frequencies and establishes a foundation for the application of OSC to colliders and other accelerator facilities.

## OSC apparatus and experiment

Our experiment uses the transit-time method of OSC, which is detailed in Fig. 1[8]. At the entrance of the cooling system, each particle passes through a pick-up undulator (PU) where it emits a short pulse of electromagnetic radiation. The beam and light are then separated using a magnetic chicane (particle bypass) that serves two functions: first, to make physical room and a temporal allowance for in-line light optics (lenses, amplifiers and delay plates), and second, to introduce a correlation between the particles' momentum deviations (errors) at the PU and their respective arrival times at the bypass exit. Finally, the kicker undulator (KU) mediates an energy exchange between the particles and their light pulses resulting in corrective energy kicks and a corresponding reduction of each particle's synchrotron (longitudinal) and betatron (transverse) oscillation amplitudes. The interaction in the KU is similar to the one that drives inverse free-electron lasers[39], devices in which an external laser field accelerates a relativistic beam as they copropagate in an undulator; however, as the radiation in OSC comes from the particles themselves, it carries information about their phase-space positions and thus can be used to produce corrections of the particles' incoherent motions.

As shown in Fig. 1b, the OSC system makes a particle's total SR loss (or its total energy change in the amplified case) highly sensitive to its energy deviation, which effectively results in a strong enhancement of the conventional SR damping rate[22]. For small phase-space deviations, the energy change is linear with the momentum deviation and gives rise to damping. With increasing momentum deviation, the single-pass OSC force oscillates, periodically reversing sign. The cooling rate, which is determined by averaging the OSC force over the particles' synchrotron and betatron motions, oscillates as well, giving rise to cooling and heating zones in phase space (Methods). The size (or acceptance) of the first cooling zone relative to the beam's root-mean-square (r.m.s.) momentum spread and r.m.s. transverse emittance (absent OSC) is termed the cooling range, and all particles within this range are effectively damped by OSC[9]. Detuning the optical delay by half a wavelength inverts the cooling and heating zones (Fig. 1b and Methods). In this

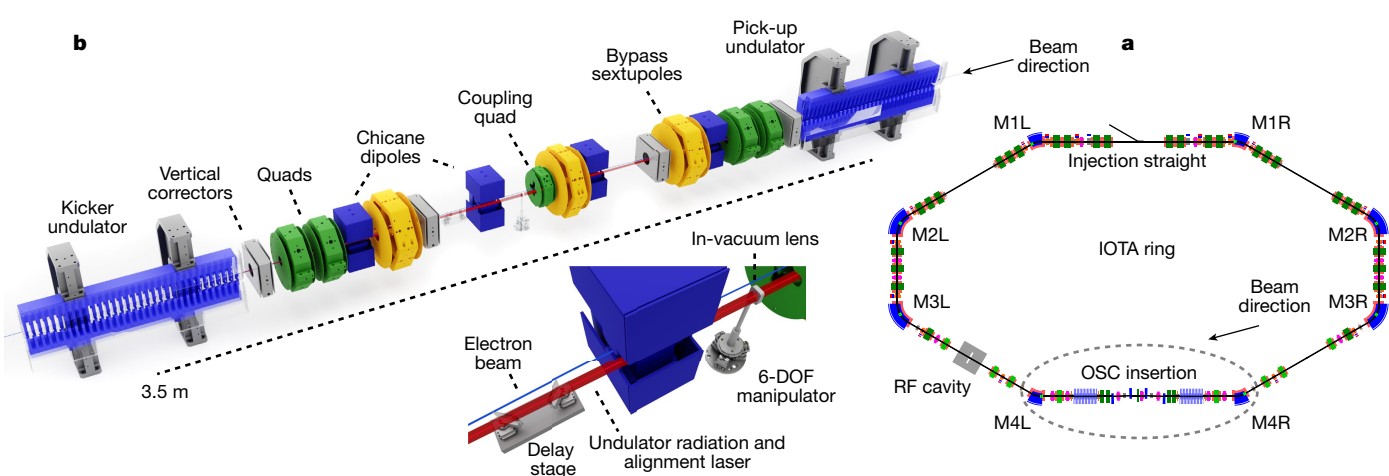

**Fig. 2 | Schematic of the IOTA OSC system. a**, Schematic of the IOTA ring and the location of the OSC insertion. **b**, Diagram of the OSC insertion including the undulators, chicane and light optics (inset). RF, radio frequency; DOF, degrees of freedom.

configuration, the small-amplitude motion becomes unstable, and the particles are antidamped to large amplitudes determined by the balance of the OSC heating and SR damping. The cooling rate and the cooling ranges described here are the essential figures of merit for OSC (Methods)[9].

The OSC cooling force can also be redistributed between the longitudinal and transverse degrees of freedom, enabling cooling in one, two or three dimensions (Methods). Lastly, the short wavelength of the radiation places stringent requirements on the alignment of the optical system and on the synchronization and stability of the bypass timing. For efficient OSC, the PU radiation and beam trajectory in the KU should be aligned to better than about 100 μm and about 100 μrad in transverse position and angle, respectively, and the bypass timing should be synchronized and stable at the subfemtosecond level[9].

The Integrable Optics Test Accelerator (IOTA), shown schematically in Fig. 2a, is a 40-m circumference, electron and proton storage ring at the Fermi National Accelerator Laboratory[10]. Table 1 presents a summary of relevant performance parameters for IOTA in the OSC configuration[9]. The OSC insertion, shown in Fig. 2b, occupies the approximately 6-m-long straight section between IOTA's M4L and M4R dipoles. The magnetic bypass uses rectangular dipoles to minimize horizontal beam focusing, ensuring that the longitudinal-to-transverse coupling is dominated by the small coupling quadrupole magnet at the centre of the bypass[9]. The PU and KU are identical electromagnetic undulators with $N_u = 16$ magnetic periods (4.84 cm each) and produce an on-axis fundamental radiation wavelength of $\lambda_r = 950$ nm for the design energy of 100 MeV (Methods). The radiation from the PU is relayed to the KU using a single in-vacuum lens with a focal length of 0.853 m at the fundamental wavelength. Although this configuration does not include optical amplification, it still produces strong cooling and enables detailed measurements of the underlying physics[9]. Before entering the KU, the light passes through a delay stage that has approximately 0.1 mm of tunable range, a closed-loop precision of about 10 nm and negligible reflection losses (Methods).

The beam's closed orbit (CO) and spatial distributions were characterized using a suite of beam-position monitors, SR monitors and a streak camera (Methods). In addition, the PU and KU radiation was monitored using two cameras at M4L that are positioned to image from different locations inside the KU. The measured positions of the

focused PU and KU radiation spots were used in conjunction with a laser-based alignment system to monitor the errors of the CO inside the undulators (Methods). The PU and KU radiation spots were spatially aligned using orthogonal CO bumps and transverse translations of the in-vacuum lens, and the delay stage was then swept through its full range until interference of the fundamental radiation was observed. After alignment, the effect of OSC on the beam distribution was apparent, and the strength of the OSC interaction was optimized using lens translations and CO bumps. After optimization, the OSC interaction was characterized using a combination of slow-delay scans and fast on–off toggles through rapid changes of the delay setting.

For the experiments described here, the OSC system was configured for three-dimensional cooling $(z, x, y)$. When the system is properly tuned, the beam's response to OSC is striking owing to OSC's dominance over SR damping. Figure 3 presents the projected beam distributions and r.m.s. sizes (Fig. 3c) for all three phase-space planes during a slow-delay scan (about 30 nm s$^{-1}$, or equivalently about $0.03\lambda_r$ s$^{-1}$) over a total range of approximately $30\lambda_r$.

**Table 1 | Design-performance parameters for IOTA OSC[9]**

| | |
|---|---|
| Design momentum, $p_0$ (MeV c$^{-1}$) | 100 |
| Revolution frequency (MHz) | 7.50 |
| Radio frequency (MHz) | 30.00 |
| Momentum compaction | $4.91\times10^{-3}$ |
| Relative r.m.s. momentum spread, $\sigma_p/p_0$[a] | $0.986\times10^{-4}$ |
| Horizontal emittance: $x$–$y$ uncoupled, $\varepsilon_0$ (nm)[a] | 0.857 |
| Total bypass delay (mm) | 0.648 |
| Nominal radiation wavelength, $\lambda_r$ (nm) | 950 |
| Maximum OSC kick per turn (meV) | 60 |
| Horizontal cooling acceptance, $\varepsilon_{max}$ (nm) | 72 |
| Longitudinal cooling acceptance, $(\Delta p/p)_{max}$ | $5.7\times10^{-4}$ |
| Bandwidth of the OSC system (THz) | 19 |
| Sum of emittance OSC rates (s$^{-1}$) | 38 |
| SR emittance damping rates, $[z,x,y]$ (s$^{-1}$) | 2.06, 0.94, 0.99 |

[a]Equilibrium values do not include the effects of OSC.

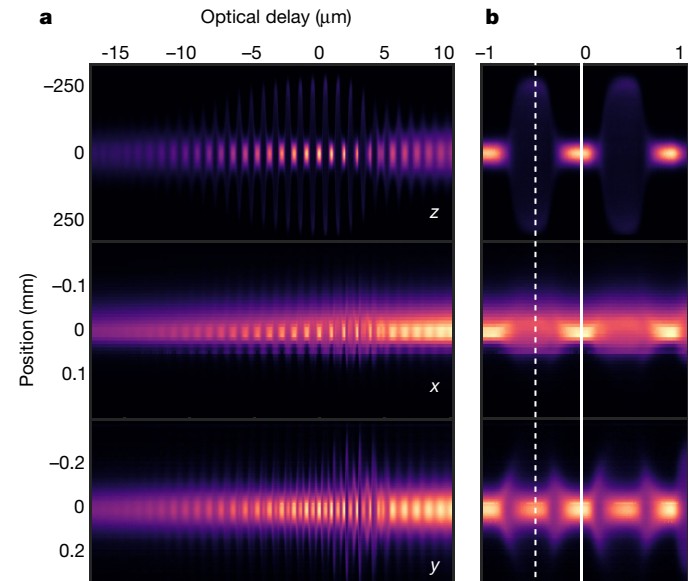

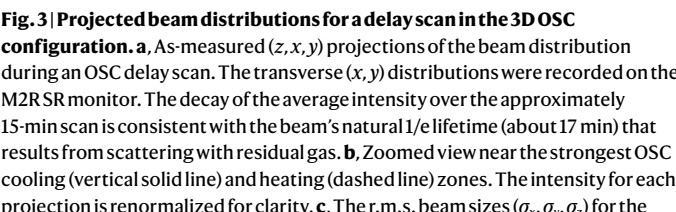

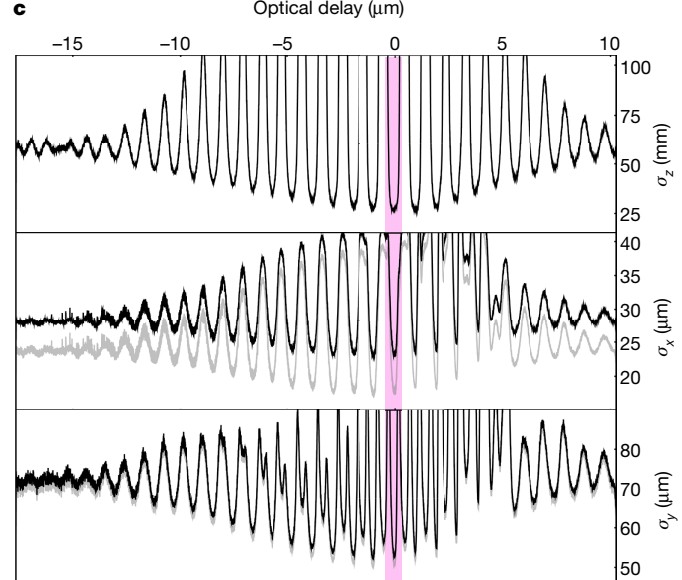

**Fig. 3 | Projected beam distributions for a delay scan in the 3D OSC configuration. a**, As-measured (*z*, *x*, *y*) projections of the beam distribution during an OSC delay scan. The transverse (*x*, *y*) distributions were recorded on the M2R SR monitor. The decay of the average intensity over the approximately 15-min scan is consistent with the beam's natural 1/e lifetime (about 17 min) that results from scattering with residual gas. **b**, Zoomed view near the strongest OSC cooling (vertical solid line) and heating (dashed line) zones. The intensity for each projection is renormalized for clarity. **c**, The r.m.s. beam sizes ($\sigma_x$, $\sigma_y$, $\sigma_z$) for the projected distributions. The fit is performed on the Gaussian core of the beam to reduce the impact of depth-of-field effects in the horizontal plane and to avoid contamination by the non-Gaussian tails that result from gas scattering. For all planes, the vertical axis is clipped at the amplitude where the core of the distribution becomes non-Gaussian owing to OSC heating. Diffraction-corrected curves are shown in grey, and the pink band marks the location of the strongest OSC cooling zone. The number of modulations in the beam size is about $2N_u$, as expected from theory[9].

The principal features of the projections can be well understood within the existing theoretical framework (Methods)[9]. At the beginning and end of the scan (high delay and low delay, respectively), the particles and light are longitudinally separated in the KU, which effectively turns off OSC and results in the equilibrium state being set by SR damping alone. As the scan proceeds, OSC alternates between cooling and heating modes with the total number of modulation periods being about $2N_u$. The OSC strength peaks after about ten periods owing to intentional overfocusing by the in-vacuum lens[9] and falls off to either side owing to the envelope function discussed in Methods. In the heating mode, large amplitudes in one plane can lead to an inversion of OSC in the other planes (Methods). This is clearly seen in Fig. 3b (white dashed line) where strong longitudinal antidamping results in cooling for the transverse planes despite the OSC system being tuned for the heating mode. The effect is less apparent for the horizontal plane than for the vertical owing to the large (dispersive) contribution of the momentum spread to the horizontal beam size at the SR monitor's location. In contrast, when OSC is weak and comparable to SR damping, as it is towards the edges of the scan (Fig. 3a), the antidamping is weak, and the cooling and heating are fully synchronized across the different planes.

The OSC cooling rates were estimated from the changes in the equilibrium beam sizes. In the fully coupled configuration (Methods), it is sufficient to consider the longitudinal distribution measured by the streak camera and the vertical distribution measured by a single SR monitor. Accordingly, the M2R SR monitor was upgraded to improve its transverse resolution (Methods). The resulting diffraction limit was small (about 15 μm) with minimal effect on the vertical beam size (Fig. 3c) and, consequently, on the inferred OSC rates (about 5%). Figure 4 presents the longitudinal and vertical distributions and their Gaussian fits for a typical fast toggle of the OSC system. The delay system is initially misaligned by about $30\lambda_r$, and the equilibrium beam distribution is set only by SR damping. At time *t* = 0, the system is moved at a rate of about $15\lambda_r \, s^{-1}$ to the strongest cooling zone. This transition is slow compared with the OSC damping times but ensures that the equilibrium distributions are taken under essentially the same conditions.

In the absence of IBS, the ratio of damping rates (with and without OSC) is inversely proportional to the ratio of the corresponding beam sizes squared; the presence of IBS reduces the difference in these beam sizes. The sample data of Figs. 3 and 4 were taken at low beam current (about 50–150 nA, or about $10^5$ particles) to reduce the impact of IBS, and a simple IBS model was used in the analysis to correct for any residual effect (Methods)[9]. Relative to SR damping alone, the equilibrium sizes correspond to an increase in the total damping rate of approximately 8.06 times and 2.94 times in the longitudinal and transverse planes, respectively (Methods). When combined into a single plane, the total amplitude cooling rate of OSC is about $9.2 \, s^{-1}$, which corresponds to a total emittance cooling rate of $18.4 \, s^{-1}$ and is about an order of magnitude larger than the longitudinal SR damping. Although not described here, cooling was also achieved with a comparable total OSC damping rate for one-dimensional (*z* only) and two-dimensional configurations (*z* and *x*). To achieve this, the IOTA ring was *x*–*y* decoupled and the longitudinal-to-transverse coupling strength was smoothly adjusted by changing the excitation of the coupling quadrupole.

The longitudinal cooling range was calculated from measurements of the beam distribution with OSC tuned for the heating mode (Methods). The observed equilibrium amplitudes are in good agreement with theoretical predictions and were calculated using two separate approaches: from the ratio of OSC to SR damping rates and directly from the streak-camera calibration. For the maximum achieved OSC rates, the calculated amplitudes coincide with one another to about 5% accuracy and, when normalized to the radiation wavenumber $k_0 = 2\pi/\lambda_r$, are approximately equal to $a_{smax} \approx 3.3$ (Methods). In Fig. 4c, the longitudinal distribution is well fit by a pure Gaussian, indicating that the beam is well within the longitudinal cooling range and there is no reduction of the damping rate for the distribution tails.

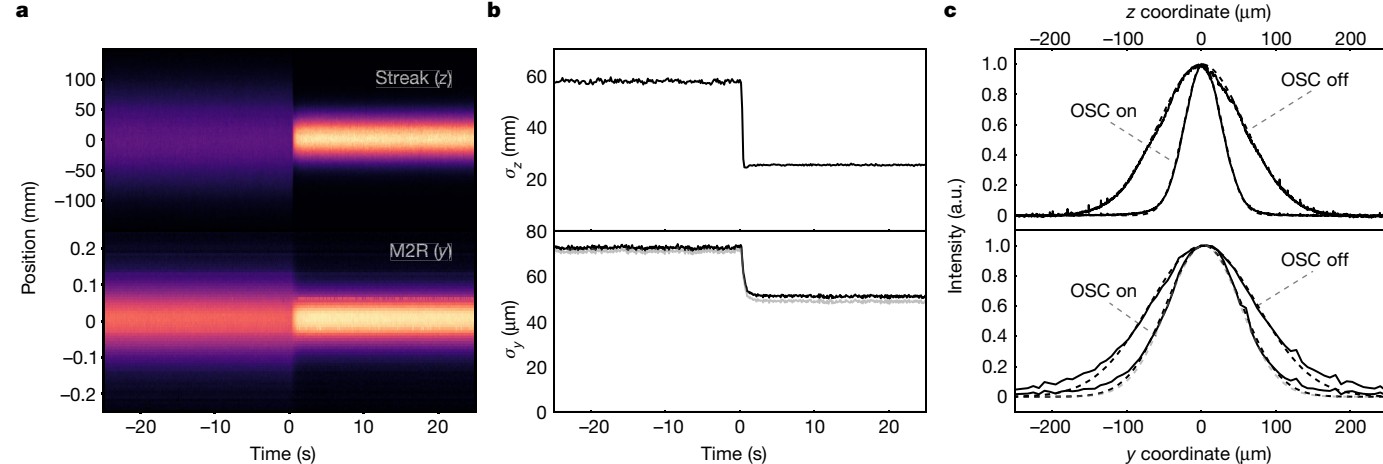

**Fig. 4 | Fast toggle of the OSC system. a,** Dependence on time of single-dimensional beam distributions in *z* (streak camera) and *y* (M2R SR monitor) during an OSC toggle. The system is initially detuned by $30\lambda_r$ and is snapped to the maximum cooling setting at $t = 0$. **b,** The r.m.s. beam sizes from Gaussian fits of the raw projections presented in **a. c,** Distributions averaged over time (solid lines) and their Gaussian fits (dotted lines) for the OSC-off and OSC-on states for the intervals of [−20, −10] s and [10, 20] s. In **b** and **c,** the M2R fits use only the central ±110 μm to reduce contamination by the non-Gaussian tails resulting from gas scattering. Diffraction-corrected curves are shown in grey, and the distributions in each case have been normalized to a peak value of one for comparison.

The transverse cooling range could not be studied in detail owing to the small transverse beam sizes and relatively weak transverse OSC, which precluded particle trapping at large transverse amplitudes; however, we note that the measured r.m.s. transverse mode emittances of the cooled beam at the OSC maximum are about 0.9 nm, which is almost 100-times smaller than the expected cooling range (Table 1) and at least about 30-times smaller than the worst-case estimate of the cooling range when reduced by the nonlinearity of longitudinal displacements at large amplitudes. Therefore, cooling-range limitations are not expected to have any role in the OSC measurements, but this conclusion still requires experimental verification.

In these results, there are a few notable deviations from expectations. The estimated total cooling rate is approximately half of the anticipated value (Table 1), which is based on detailed simulations of the undulator radiation[9]. Also, the ratio of the measured OSC rates (longitudinal to sum of transverse) in the experiment was 1:0.34, whereas 1:1.03 is expected[9]. The likely sources of these discrepancies are discussed in Methods.

## Conclusions

We have experimentally demonstrated optical stochastic cooling. This constitutes the realization of a stochastic beam-cooling technique in the terahertz-bandwidth regime and represents an increase in bandwidth of about 2,000 times over conventional SC systems. In addition, we have successfully demonstrated a coupling scheme for sharing the cooling force with all degrees of freedom, which is applicable to other cooling concepts as well. Another important technical outcome of the experiment is that the beam and its radiation were effectively synchronized and stabilized to better than a quarter of the radiation wavelength (<250 nm) over the length of the OSC section (about 3 m). These results provide an important validation of the essential OSC physics and technology and open the way to experiments that include high-gain optical amplification and advanced system architectures. For example, the next phase of the IOTA OSC programme is underway and targets the development of an amplified OSC system with about 4–6 mm of delay, an optical power gain of >30 dB and the flexibility to explore advanced concepts that will broaden the applicability of OSC, such as transverse optical sampling[40]. The successful demonstration of this amplified system would provide the foundation necessary for engineering operational, high-gain OSC systems for colliders and other accelerator facilities and may open capabilities for synchrotron light sources. These may include OSC systems for direct cooling of hadron beams, secondary cooling of stored high-intensity electron beams for ring-based electron coolers and flexible OSC systems for enhanced SR damping.

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

# Methods

## Measurement with synchrotron-radiation monitors

IOTA is equipped with synchrotron-radiation monitors at each main dipole. These stations, which use Blackfly-PGE-23S6M-C complementary metal−oxide−semiconductor (CMOS) cameras and have approximately unity magnification, are used to record direct transverse images of the beam distribution. For the OSC experiments, the M2R station was upgraded for higher resolution using two hardware improvements: the rejection of vertically polarized light (Thorlabs PBSW-405) and the use of a narrowband filter (Thorlabs FBH405-10). Although the emitted vertically polarized light is weak, it increases the diffraction-limited spot size by about 20%. The wavelength of the narrowband filter was made as short as possible while maintaining a high quantum efficiency in the camera's sensor. The filter reduced the diffraction contribution of long-wavelength radiation and the contribution of lens chromaticity from short-wavelength radiation. Calculations indicate that these measures reduced the diffraction contribution by almost a factor of two. To focus the beam image, the longitudinal position of each camera was adjusted to minimize the measured beam size; preference was given to the vertical size as depth-of-field effects are more substantial in the horizontal size owing to the horizontal curvature of the beam trajectory. 'Hot' pixels in the M2R projections were selected using a peak-detection algorithm and replaced by the average of their nearest neighbours. The diffraction corrections to the measured beam sizes were determined experimentally for the M2R and M1L measurement stations by fitting a single correction to the cooling, heating and no-OSC configurations for various beam currents. Examples of the measured and corrected beam sizes are shown for sample data in Extended Data Fig. 1. The experimentally determined corrections are close to the theoretical estimates of 15 μm and 31 μm for M2R and M1L, respectively.

## Alignment of the OSC system

The OSC insertion is equipped with a laser-based alignment system for supporting the alignment of diagnostic systems and manipulation of the CO for OSC tuning. The system comprises a helium−neon laser (632.8 nm) with its axis aligned through two surveyed pinholes at either end of the OSC insertion. The transverse positioning errors of the pinholes are approximately ±50 μm. A set of air-side matching optics is used to focus the alignment laser to the centre of the PU. The laser then relays through the in-vacuum optics and all downstream diagnostic lines to produce submillimetre images on the undulator radiation (UR) cameras described below. Before the OSC experiments, the CO was corrected to a few-hundred micrometres using quad-centring, and the lattice functions were corrected to several percent using standard techniques[40,41].

## In-vacuum light optics

The in-vacuum lens and delay plates are fabricated from CORNING-HPFS-7980, which was chosen primarily for its low-group-velocity dispersion in the wavelength range of interest. The lens is anti-reflection coated for the fundamental band of the UR (950–1,400 nm) and has a clear aperture of about 13 mm, which corresponds to an acceptance angle for the PU radiation of about 3.5 mrad. The lens position can be adjusted in six degrees of freedom (<±10 nm in positions and <±15 μrad in angles) using an in-vacuum, piezo-electric manipulator (Smaract Smarpod 70.42) operating in a closed-loop mode. The delay plates have a central thickness of 250 μm and the typical variation over the 25-mm plates was measured via Haidinger interferometry to be about 100 nm. The nominal orientation of the delay plates is near the Brewster angle to reduce reflection losses of the PU light. The delay system uses two closed-loop, rotary piezo stages (Smaract SR-2013) to provide independent rotation of the two delay plates. The delay can be tuned over a full range of approximately 0.1 mm with a precision of about 10 nm. Although the absolute angles (that is, relative to the OSC alignment axis) of the plates are not known, a delay model can be fit to the periodicity of the OSC cooling force for a continuous-rate angular scan.

## Undulator-radiation diagnostics

Imaging of the radiation from the PU and KU provides important diagnostic capabilities. The undulators produce an on-axis radiation wavelength given by $\lambda_r = l_u(1 + K^2/2)/2n\gamma^2$, where $l_u$ is the undulator period, $n$ is the harmonic of the radiation, $\gamma$ is the Lorentz factor, $K = q_e B l_u/2\pi m_e c$ is the undulator parameter, $q_e$ is the electron charge, $B$ is the peak on-axis magnetic field, $m_e$ is the electron rest mass and $c$ is the speed of light. A lightbox located on the M4L dipole contains all diagnostic systems for the KU and PU radiation. Two Blackfly-PGE-23S6M-C CMOS cameras are used in combination with a filter wheel to image the fundamental, second or third harmonic from the KU and PU. These UR cameras are located at two separate image planes corresponding to different locations inside the KU. As the in-vacuum lens is in a 2$f$-relay configuration, the PU light is mapped into the KU with an approximately negative-identity transformation. The imaging system then produces a single, relatively sharp image of the beam, for both the PU and KU, from the corresponding source plane. In conjunction with the alignment system, these images can be used to estimate the trajectory errors of the closed orbit in both undulators. To validate the concept, the propagation of realistic UR through the entire imaging system was performed in the Synchrotron Radiation Workshop[42]. In practice, the simultaneous imaging of the KU and PU radiation from the same source plane in the KU enables straightforward, rough alignment of the KU closed orbit with the PU radiation, which along with longitudinal alignment is the principal requirement for establishing the OSC interaction. The cameras have sufficient infrared quantum efficiency to directly image the interference of the fundamental radiation, which indicates successful longitudinal alignment.

## Longitudinal-beam measurements

A Hamamatsu model C5680 dual-sweep streak camera with a synchroscan (M5675) vertical deflection unit was used to measure the beam's longitudinal distribution during the OSC experiments. A Blackfly-PGE-23S6M-C CMOS camera was used as the detector element, and the system was installed above the M3R dipole. A 50/50 non-polarizing beam splitter was used to direct half of the SR from the existing M3R SR beam-position monitor to the entrance slit of the streak camera. An external clock generator was phase locked to IOTA's fourth-harmonic radio frequency (30 MHz) and used to drive the streak camera's sweep at the eleventh harmonic (82.5 MHz) of the beam's circulation frequency (7.5 MHz). To calibrate the streak-camera image (picoseconds per pixel), IOTA's radio-frequency voltage was first calibrated using a wall current monitor to measure shifts in the synchronous phase of the beam for different voltage settings. Measurements of the synchrotron frequency (by resonant excitation of the beam) were then made as a function of voltage setting, which yielded a small correction factor for the momentum compaction (+15%). This value is highly sensitive to focusing errors in the low-emittance lattice designed for the OSC studies. Finally, the streak camera's calibration factor was determined by fitting the measured longitudinal-beam position as a function of voltage. A slight nonlinearity was observed at the edge of the system's field of view. It was corrected by making the beam's longitudinal distribution symmetric relative to its centre for OSC in the heating mode, where the amplitudes were largest and the bunch lengths barely fit within the field of view. Extended Data Fig. 2 presents an example of the longitudinal distribution before and after correction.

## Specialized power systems

The chicane dipoles are powered in pairs using special current regulators (BiRa Systems PCRC) with a ripple-plus-noise at the $1 \times 10^{-5}$ level (r.m.s.) and a long-term stability of a few parts per million. This ensures

that the nominal phase of the OSC force is stable as the beam particles sample it over many turns. It is noted that regulation at the mid $10^{-5}$ level corresponds to effective momentum errors comparable to the beam's natural momentum spread. The regulation of the main power supply that feeds IOTA's dipoles was also improved to comparable stability. This was required for IOTA because, as an electron synchrotron with fixed radio frequency, the beam energy is directly related to the magnetic field in the ring's dipoles; therefore, variations of the bending field result in variations of the particle delay.

## Coupling of the phase-space planes

Energy exchange between the particles and their PU radiation fields in the KU is a longitudinal effect; however, as described in the next section, the presence of dispersion in the undulators can be used to couple the cooling force into the transverse phase planes. In the system reported here, this coupling (longitudinal to horizontal) is smoothly adjustable by excitation of a single quadrupole in the centre of the OSC bypass. Operation of the storage ring on a transverse coupling resonance, in our case a difference resonance with betatron tunes (that is, number of betatron oscillations per revolution) of $Q_x = 5.42$ and $Q_y = 2.42$, splits the beam emittance and cooling and heating between the horizontal and vertical planes. This combination of bypass and lattice coupling enables full three-dimensional cooling of the beam using OSC. To couple the lattice, the ring optics were corrected to minimize the split between the fractional part of the betatron tunes ($\Delta Q < 0.005$), and then strong transverse coupling was introduced by excitation of a single skew quadrupole in a region with zero dispersion.

## Analysis of OSC rates and ranges

For the derivation of the OSC cooling rates, we refer the reader to ref. [9]. Here we only summarize the major results needed for analysis of the experimental data. For relatively small momentum deviation, the longitudinal kick experienced by a particle can be approximated as

$$\delta p/p = \kappa u(s) \sin(k_0 s), \tag{1}$$

where $\kappa$ is the maximum kick value, $k_0 = 2\pi/\lambda_r$ is the radiation wavenumber, $s$ is the particle's longitudinal displacement on the way from the PU to the KU relative to the reference particle, which obtains zero kick, and $u(s)$ is an envelope function, with $u(0) = 1$ and $u(s) = 0$ for $|s| > N_u \lambda_r$, that accounts for the bandwidth of the integrated system. The effects of the envelope function are observed in Fig. 3c. In the linear approximation, one can write

$$s = M_{51} x + M_{52} \theta_x + M_{56}(\Delta p/p), \tag{2}$$

where $M_{sn}$ are the elements of $6 \times 6$ transfer matrix from pick-up to kicker, and $x$, $\theta_x$ and $\Delta p/p$ are the particle coordinate, angle and relative momentum deviation in the PU centre. To find the longitudinal cooling rate for small-amplitude motion, we leave only the linear term in $ks$ in equation (1) and set $u(s) = 1$. The longitudinal cooling rate is straightforwardly obtained as

$$\lambda_s = \frac{\kappa}{2} f_0 k_0 (M_{51} D + M_{52} D' + M_{56}), \tag{3}$$

with $D$ and $D' = dD/ds$ being the ring dispersion and its longitudinal derivative at the PU. Here we also include that for pure longitudinal motion $x = D(\Delta p/p)$ and $\theta_x = D'(\Delta p/p)$. Then, using symplectic perturbation theory and the rate-sum theorem[43], one obtains that the sum of cooling rates (in amplitude) is equal to the longitudinal cooling rate in the absence of $x$–$s$ coupling:

$$\lambda_1 + \lambda_2 + \lambda_s = \frac{\kappa}{2} f_0 k_0 M_{56}, \tag{4}$$

where $\lambda_1$ and $\lambda_2$ are the cooling rates of the two betatron modes, $\lambda_s$ is the cooling rate of longitudinal motion and $f_0$ is the revolution frequency in the storage ring.

In the general case of arbitrary $x$–$y$ coupling, the cooling rates for the transverse modes have lengthy expressions; however, for the case of operation at the coupling resonance the cooling rates of two transverse modes are equal and have a compact representation. In this case, combining equations (3) and (4), one obtains

$$\lambda_1 = \lambda_2 = -\frac{\kappa}{4} f_0 k_0 (M_{51} D + M_{52} D'). \tag{5}$$

The harmonic dependence of the cooling force on momentum deviation, presented in equation (1), results in a reduction of the cooling rates with increasing amplitude. Averaging over betatron (transverse) and synchrotron (longitudinal) oscillations yields the dependence of cooling rates on the particle amplitudes:

$$\begin{bmatrix} \lambda_1(a_1, a_2, a_s) \\ \lambda_2(a_1, a_2, a_s) \\ \lambda_s(a_1, a_2, a_s) \end{bmatrix} = 2 \begin{bmatrix} \lambda_1 J_1(a_1) J_0(a_2) J_0(a_s)/a_1 \\ \lambda_2 J_0(a_1) J_1(a_2) J_0(a_s)/a_2 \\ \lambda_s J_0(a_1) J_0(a_2) J_1(a_s)/a_s \end{bmatrix}, \tag{6}$$

where $J_n$ is the $n$th-order Bessel's function of the first kind, $a_1$, $a_2$ and $a_s$ are the dimensionless amplitudes of the particle's longitudinal displacement in the kicker related to the oscillations in the corresponding plane. Expressed in units of the phase of the electromagnetic field they are given by

$$a_1 = k_0 \sqrt{\varepsilon_1 (\beta_{1x} M_{51}^2 - 2\alpha_{1x} M_{51} M_{52} + ((1-u)^2 + \alpha_{1x}^2) M_{52}^2/\beta_{1x})},$$

$$a_2 = k_0 \sqrt{\varepsilon_2 (\beta_{2x} M_{51}^2 - 2\alpha_{2x} M_{51} M_{52} + (u^2 + \alpha_{2x}^2) M_{52}^2/\beta_{2x})},$$

$$a_s = k_0 (M_{51} D + M_{52} D' + M_{56}) (\Delta p/p)_{max}, \tag{7}$$

where $(\Delta p/p)_{max}$ is the amplitude of the synchrotron motion, $\varepsilon_1$ and $\varepsilon_2$ are the generalized Courant–Snyder invariants (single-particle emittances), and $\beta_{1x}, \beta_{2x}, \alpha_{1x}, \alpha_{2x}$ and $u$ are the four-dimensional Twiss parameters defined in Section 2.2.5 of ref. [43]. The cooling rates in equation (6) oscillate with the particle amplitudes, and as a result, particles may be trapped at large amplitudes by the OSC force. The requirement to have simultaneous damping for all degrees of freedom determines the cooling acceptances so that $a_i \lesssim \mu_{01} \approx 2.405$, $i = 1, 2, s$; where $\mu_{01}$ is the first root of Bessel function $J_0(x)$. If oscillations happen in only one degree of freedom, then the cooling range is larger: $a_i \lesssim \mu_{11} \approx 3.83$, where $\mu_{11}$ is the first non-zero root of Bessel function $J_1(x)$.

Although the small-amplitude OSC rates greatly exceed the SR cooling rates, accounting for SR is important to understand the observed beam behaviour. In this case the total cooling rate for the $n$th degree of freedom is:

$$\lambda_n = \lambda_{nSR} (1 + 2R_{n\tau} J_1(a_n) J_0(a_m) J_0(a_k)/a_n), \, n \neq m \neq k, \tag{8}$$

where $R_{n\tau}$ is the ratio of the small-amplitude OSC rate to the SR cooling rate for the $n$th degree of freedom, $m$ and $k$ are the other degrees of freedom, and the label $\tau$ is used to indicate a ratio of cooling rates. In our measurements with the antidamping OSC phase, the dimensionless amplitudes of betatron motion are much smaller than one. For longitudinal OSC in the antidamping mode, one then obtains the dependence of the longitudinal cooling rate on the dimensionless amplitude of the synchrotron motion as:

$$\lambda_s = \lambda_{sSR} (1 - 2R_{s\tau} J_1(a_s)/a_s). \tag{9}$$

Consequently, the equilibrium amplitude is determined by the following equation: $a_s = 2R_{sr}J_1(a_s)$. Extended Data Fig. 3 presents the dependence of the longitudinal cooling rates for OSC in the damping and antidamping modes for the measured parameters of OSC. For both modes, there is only one equilibrium point: $a_s = 0$ for the damping mode, and $a_s = 3.273$ for the antidamping mode.

For very small beam current, where IBS is negligible, the r.m.s. emittance growth rate of small-amplitude motion is determined by the following equation:

$$\frac{d\varepsilon_n}{dt} = -2\lambda_n\varepsilon_n + B_n, \ n = 1, 2, s. \tag{10}$$

Here $B_n$ is the diffusion driven by fluctuations from SR emission and scattering from residual gas molecules, and thus does not depend on the beam parameters. In equilibrium, equation (10) determines the cooling rate, $\lambda_n = B_n/2\varepsilon_n$, and a straightforward way to compute the cooling rate from the ratio of r.m.s. beam sizes with ($\sigma_n$) and without ($\sigma_{n0}$) OSC:

$$\frac{\lambda_n}{\lambda_{n0}} = (\sigma_{n0}/\sigma_n)^2, \tag{11}$$

where $\lambda_{n0}$ is the damping rate in the absence of OSC. Although all reported OSC measurements were done with a small beam current (about 50–150 nA), for a large fraction of the measurements IBS was not negligible; therefore, we use a simplified IBS model to calculate corrections to the cooling rates that are largely independent of exact beam parameters[44]. We add the IBS term to the right-hand side of equation (10):

$$\frac{d\varepsilon_n}{dt} = -2\lambda_n\varepsilon_n + B_n + A_n\frac{N}{\varepsilon_\perp^{3/2}\sqrt{\varepsilon_s}}, \tag{12}$$

where $\varepsilon_\perp = \varepsilon_1 = \varepsilon_2$ is the r.m.s. transverse emittance, $\varepsilon_s$ is the r.m.s. longitudinal emittance and the constant $A_n$ is determined from the measurements.

To find $A_n$, we use the fact that the r.m.s. beam sizes are different at the beginning and at the end of OSC sweep, which continues for about 1,000 s. For the measurement presented in Fig. 3, the measured beam lifetime of 17 min yields the ratio of beam currents at the beginning and at the end of the measurements to be $R_N = 2.5$. With some algebraic manipulations, one can express the ratio of the measured beam sizes at the sweep end, $\sigma_{n2}$, to the beam sizes that would be measured in the absence of IBS, $\sigma_{n0}$, through the ratios of other measured parameters as:

$$R_v \equiv \left(\frac{\sigma_{v0}}{\sigma_{v2}}\right)^2 = \left(\frac{\sigma_{v1}}{\sigma_{v2}}\right)^2 - \frac{(\sigma_{v1}/\sigma_{v2})^2 - 1}{1 - (\sigma_{v1}/\sigma_{v2})^3(\sigma_{s1}/\sigma_{s2})/R_N}. \tag{13}$$

Here $(\sigma_{v1}/\sigma_{v2})$ is the ratio of the initial and final vertical beam size, $(\sigma_{s1}/\sigma_{s2})$ is the same measure for the bunch lengths, the emittances of both transverse modes are taken to be equal and we have used $\varepsilon_{\perp 1}/\varepsilon_{\perp 2} = (\sigma_{v1}/\sigma_{v2})^2$. For these experiments, the approximate ratios of the beam sizes (before and after sweep) are: $(\sigma_{v1}/\sigma_{v2}) = 1.09$ and $(\sigma_{s1}/\sigma_{s2}) = 1.1$; that yields $(\sigma_{v0}/\sigma_{v2})^2 = 0.745$.

In the next step, we find the ratio of cooling rates with and without OSC. Similar manipulations with equation (12) yield an improved version of equation (11):

$$\frac{\lambda_{vOSC}}{\lambda_{vSR}} = \left(\frac{\sigma_{vSR}}{\sigma_{vOSC}}\right)^2\frac{1}{R_{IBS}(1-R_v)+R_v}, R_{IBS} = \left(\left(\frac{\sigma_{vSR}}{\sigma_{vOSC}}\right)^3\frac{\sigma_{sSR}}{\sigma_{sOSC}}\right)^{-1} \tag{14}$$

Here $\sigma_{vSR}/\sigma_{vOSC}$ is the ratio of the measured beam sizes without and with OSC, respectively, $\sigma_{sSR}/\sigma_{sOSC}$ is the same for the bunch lengths. For the measurements presented here, $\sigma_{vSR}/\sigma_{vOSC} = 1.51$ and $\sigma_{sSR}/\sigma_{sOSC} = 2.5$,

which yields $\lambda_{vOSC}/\lambda_{vSR} = 2.94$. Similar calculations for the longitudinal cooling yield $\lambda_{sOSC}/\lambda_{sSR} = 8.06$. The typical day-to-day variability of the rates was around the 10% level owing to variations in the overall tuning and alignment of the OSC system and CO; however, the performance was stable during any given operations session with only infrequent, minor tuning required.

## Gas scattering and ring acceptance

Fits to the vertical distributions in Fig. 4 reveal two features of note: (1) the equilibrium size in the OSC-off case is approximately two-times larger than anticipated[9] and (2) the distributions have non-Gaussian tails for both cases, with and without OSC. Both observations are consistent with scattering from residual gas molecules. The average vacuum pressure in the storage ring, which is estimated from the beam size increase, is about $3.7 \times 10^{-8}$ torr of atomic hydrogen equivalent and coincides with the vacuum estimate of the previous IOTA run to within about 15% accuracy[45]. The vertical acceptance of the storage ring is smaller than the horizontal acceptance and is estimated from the beam lifetime to be about 3 μm, which is about 50% of the design value and exceeds the cooling range (Table 1) by a factor of about 40.

## Cooling-rate and coupling discrepancies

One likely source of the apparent cooling-rate discrepancy is a reduced physical aperture for the PU light owing to misalignments of the vacuum chambers and the beam. Preliminary simulations based on three-dimensional scans of the integrated apparatus suggest that on the order of 30–40% of the difference might be accounted for in this way. Other potential sources may include nonlinearities in the bypass mapping, the possibility of distorted CO trajectories in the undulators owing to saturation in the steel poles, and reduced energy exchange owing to the finite radiation spot size in the KU, which is exacerbated by the beam's increased size owing to residual gas scattering.

Regarding longitudinal-to-transverse coupling, the measured ratio (1:0.34) would correspond to excitation of the coupling quadrupole at approximately half its nominal strength. This suggests the presence of additional coupling terms in the bypass and/or deviations of the lattice optics functions from the model; however, in two-dimensional OSC experiments, which are not reported here, the coupling-quad excitation was doubled and a coupling ratio closer to unity was achieved.

## Data availability

The datasets for the reported experiments are available in the Zenodo repository at https://doi.org/10.5281/zenodo.6578557.

## Code availability

Code supporting the processing and analysis of the reported data is available in the Zenodo repository at https://doi.org/10.5281/zenodo.6578557.

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

**Acknowledgements** We thank D. Frank, M. Obrycki, R. Espinoza, N. Eddy and J. You for technical and hardware support; B. Cathey for operations support; and M. Zolotorev, M. Andorf, A. Lumpkin, J. Wurtele, A. Charman and G. Penn for discussions. This manuscript has been authored by the Fermi Research Alliance, LLC under contract number DE-AC02-

07CH11359 with the US Department of Energy Office of Science, Office of High Energy Physics. This work was also supported by the US Department of Energy contract Nnumber DE-SC0018656 with Northern Illinois University and by the US National Science Foundation under award PHY-1549132, the Center for Bright Beams.

**Author contributions** V.L. initiated the IOTA OSC programme, led the OSC conceptual design and data analysis, and supported the experimental measurements. J.J. led the design, simulation, integration and experimental operations of the OSC apparatus and diagnostics systems, and supported the OSC conceptual design and data analysis. A.R. supported the OSC conceptual design, hardware development and experimental measurements, and led the commissioning of the OSC lattices. A.V. and J.S. supported operations of the IOTA ring. A.V., J.S., A.R., G.S. and D.B. supported the integration of the OSC hardware. D.B. supported the integration and operation of the OSC motion systems. J.R. supported the conceptual design and the simulation and development of the OSC light optics and diagnostics. D.E. supported the operation of diagnostic systems. K.C. supported integration and operation of the RF and power systems. H.P. supported the design, modelling and fabrication of the OSC vacuum systems. G.S. and I.L. supported the OSC measurements and diagnostic systems. A.D. and P.P. supported the development of the optical delay system. S.N. and A.V. initiated the IOTA research programme, led the IOTA ring design and construction, and provided programmatic guidance and support. S.C. provided technical and academic guidance, hardware support and senior mentorship. All authors participated in group discussions that helped guide various aspects of the OSC programme. The manuscript was written by J.J. and V.L. All authors contributed to the editing of the manuscript.

**Competing interests** The authors declare no competing interests.

**Additional information**
**Correspondence and requests for materials** should be addressed to J. Jarvis or V. Lebedev.

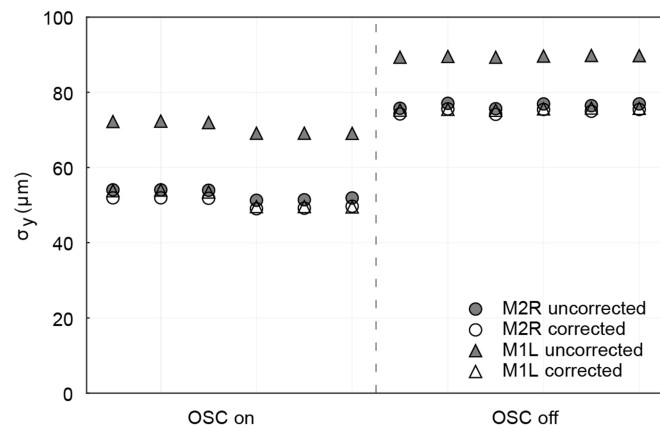

**Extended Data Fig. 1 | Diffraction correction for SR monitors.** Examples of measured rms vertical beam sizes with and without OSC for M2R (circles) and M1L (triangles) synchrotron-radiation monitors; filled - uncorrected measurements; unfilled - measurements corrected for diffraction as $\sqrt{\sigma^2 - D^2}$. The diffraction corrections are: D = 15 μm for the M2R monitor and D = 31 μm for the M1L monitor. Data for the M1L monitor were scaled to the M2R location using the ratio of the M2R to M1L beta functions. The diffraction correction to the M1L size was applied before scaling of the size to the M2R location.

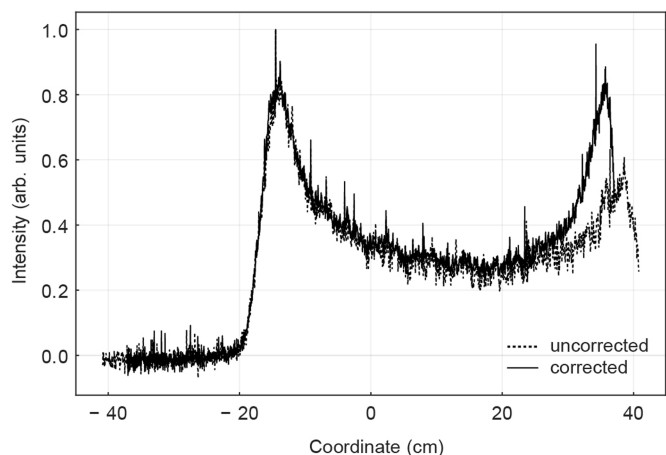

**Extended Data Fig. 2 | Correction of nonlinearity in streak camera image.**
Longitudinal distribution before (dotted) and after (solid) correction of the
streak-camera nonlinearity for the OSC in the antidamping mode.

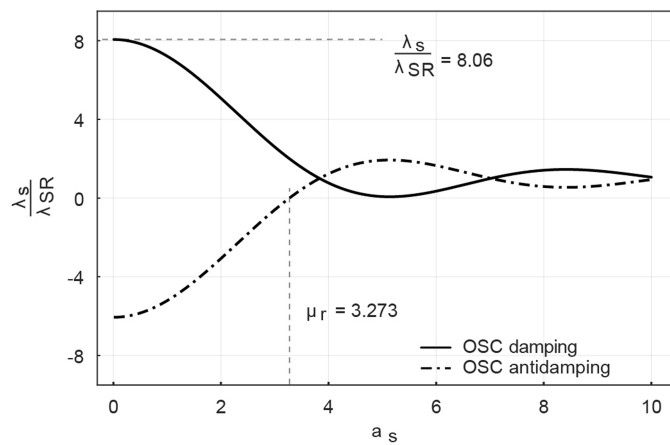

**Extended Data Fig. 3 | Computed dependence of damping rate on synchrotron amplitude for the measured OSC performance.** Total longitudinal damping (solid) and antidamping (dashed) rates as a function of normalized synchrotron amplitude. The rates are normalized to the strength of the longitudinal synchrotron-radiation damping.