## [Peer Review File · Nature]

Manuscript Title: Experimental Demonstration of Optical Stochastic Cooling

Reviewer Comments & Author Rebuttals

Reviewer Reports on the Initial Version:

Referees' comments:

Referee #1 (Remarks to the Author):

Review of Jarvis et. al.

Key results:

This paper provides convincing experimental evidence of optical stochastic cooling.

This is the first instance of ultra wideband stochastic cooling in particle accelerators and it deserves publication in Nature.

The paper begins with a discussion of beam cooling in general and then proceed to the particulars of this effort.

The discussion proceeds and culminates in figures 3 and 4, which I find compelling. You clearly did it!

Validity:

The arguments are convincing and the data are clean. There are no issues with validity.

Originality and significance:

As mentioned under key results this is new and highly significant for the physics of beams.

It has import for nuclear and high energy physics and

I would bet that some statistical mechanics experts will be interested too.

Data & methodology:

The Methods section does a good overview of the relevant calculations and details the experimental technique.

I don't see any problems here.

Suggested improvements:

I do have some issues with figure 1b. In the caption you mention that the grey section corresponds to the rms momentum

spread in the experiment and it is clearly in the cooling region. If the horizontal axis is the peak momentum deviation, during synchrotron oscillation then there is certainly net cooling out to about $\pm 1.2e-3$,

where the energy loss per turn is the same as at $dp/p=0$. In lines 113 and 114 you mention that inverting heating and cooling

requires a half wavelength shift. A sentence or two describing this would be helpful. The statement about possible heating in the pink regions is mysterious,

is that because you redistribute the cooling between dimensions? If not then what?

Also, should you really call your vertical axis SR loss per turn? Anyway, this is an important figure.

Please strive to make it clear.

The references are good and the abstract is clear and accessible. I find the conclusions to be solid.

All in all I find this to be an excellent paper.

Referee #2 (Remarks to the Author):

The article reports the first demonstration of optical stochastic cooling. This method has been proposed almost 30 years ago as an extension of the established stochastic cooling in the microwave regime to optical frequencies which promise larger system bandwidth and consequently faster cooling. The availability of optical stochastic cooling could have large impetus on planned accelerator projects and improve the performance and beam quality of future high energy accelerator rings. The presented results are certainly fundamental for further studies aiming at the development of the optical stochastic cooling method for high energy accelerator projects. The manuscript reports results which are of large importance for experts in the field of accelerators and beam cooling, but it is written in a style which addresses a broad readership. The precursory and related activities in the field are properly cited.

The report is written clearly and presents very convincing experimental data which supports the claim of the observation of optical stochastic cooling. So far, the system was only operated in a mode without amplification of the optical system, in contrast to the original proposal. However, the experiments have demonstrated the possibility to control all system parameters with the required precision and stability. The addition of optical amplification can be considered an engineering problem for the future. The basic issues of the original idea have been addressed and solved with good control of the system in this proof of principle experiment. With respect to the presented results, I recommend to indicate in Figure 1, that the experiment did only have light optics, but no amplifier.

The authors describe their experimental methods and observations in a clear way, the figures prove the claim of the observation of cooling and are of good quality. The methods to prepare the system for cooling and the methods to detect the cooling are thoroughly described. The theoretical background for analysis of the measurements is described in other publications, but shortly summarized in the methods section. For a first experiment of optical stochastic cooling the agreement between theoretical calculations and experimental results is good, confirming the excellent mastering of the experimental equipment.

One aspect of their experiment might deserve a more detailed description. The authors report coupling of orthogonal subspaces of the 6-dimensional phase space. This technique is not considered in the original proposal and is only shortly mentioned in the manuscript. The method of coupling and the observation and control of coupling could be described in a chapter in the methods section in more detail.

In equation (14) the formula describes the ratio of damping times, although the text states rates. There is no definition of the damping time in the manuscript. This part should be modified to avoid the use of damping time or the damping time should be defined.

I recommend to consider the proposed modifications. The manuscript is already of good quality and the modifications are only intended to clarify details.

In general, the manuscript describes excellent experimental results which can be of big importance for strategies to improve accelerator systems, particularly high energy colliders and low emittance, high brilliance light sources. The results have relevance both for accelerator designers and users of the accelerators. The description is appropriate for a broad readership.

In conclusion, I strongly recommend the manuscript for publication in Nature.

Some minor comments and corrections:

line 42-44: for colliders, I would mention the increase of luminosity by shrinking the emittance in

the course of cooling

line 400: 'the bunch length barely fit' -> 'the bunch lengths barely fit' or 'the bunch length barely fits'

equation (12): ϵ_z should be ϵ_s

In the reference list I found misspelling of author names:

ref. 22, 43, 44: Zimmermann

ref. 29 Brennan

Author Rebuttals to Initial Comments:

RESPONSE TO REFEREES:

Referee comments in italics

*Our responses in blue.

Referee #1:**Key results:**

This paper provides convincing experimental evidence of optical stochastic cooling.

This is the first instance of ultra wideband stochastic cooling in particle accelerators and it deserves publication in Nature.

The paper begins with a discussion of beam cooling in general and then proceed to the particulars of this effort. The discussion proceeds and culminates in figures 3 and 4, which I find compelling. You clearly did it!

Validity:

The arguments are convincing and the data are clean. There are no issues with validity.

Originality and significance:

As mentioned under key results this is new and highly significant for the physics of beams. It has import for nuclear and high energy physics and I would bet that some statistical mechanics experts will be interested too.

Data & methodology:

The Methods section does a good overview of the relevant calculations and details the experimental technique. I don't see any problems here.

Suggested improvements:

I do have some issues with figure 1b. In the caption you mention that the grey section corresponds to the rms momentum spread in the experiment and it is clearly in the cooling region. If the horizontal axis is the peak momentum deviation, during synchrotron oscillation then there is certainly net cooling out to about +/- 1.2e-3, where the energy loss per turn is the same as at dp/p=0. In lines 113 and 114 you mention that inverting heating and cooling requires

a half wavelength shift. A sentence or two describing this would be helpful. The statement about possible heating in the pink regions is mysterious, is that because you redistribute the cooling between dimensions? If not then what? Also, should you really call your vertical axis SR loss per turn? Anyway, this is an important figure. Please strive to make it clear.

We have revised Figure 1b for clarity per the referee's helpful comments. Specifically, we have removed the conceptual depiction of heating at high amplitudes and the corresponding text as the discussion is better left to the Methods section where the averaging over synchrotron and betatron oscillations, the equilibrium amplitudes and other details are discussed in detail. Since the referee pointed to the comment on inversion of the heating/cooling zones, we have added a curve to Figure 1b that shows the energy loss (vs. momentum deviation) for the heating mode, where the delay system is detuned by half an optical wavelength ($\pm \pi$ in optical phase). This illustration leads nicely into the results section, and we hope that it provides extra clarity for the reader. We have also modified the plot labels and the figure caption.

The references are good and the abstract is clear and accessible. I find the conclusions to be solid. All in all I find this to be an excellent paper.

We greatly appreciate the referee's supportive remarks.

Referee #2 (Remarks to the Author):

The article reports the first demonstration of optical stochastic cooling. This method has been proposed almost 30 years ago as an extension of the established stochastic cooling in the microwave regime to optical frequencies which promise larger system bandwidth and consequently faster cooling. The availability of optical stochastic cooling could have large impetus on planned accelerator projects and improve the performance and beam quality of future high energy accelerator rings. The presented results are certainly fundamental for further studies aiming at the development of the optical stochastic cooling method for high energy accelerator projects. The manuscript reports results which are of large importance for experts in the field of accelerators and beam cooling, but it is written in a style which addresses a broad readership. The precursory and related activities in the field are properly cited.

The report is written clearly and presents very convincing experimental data which supports the claim of the observation of optical stochastic cooling. So far, the system was only operated in a mode without amplification of the optical system, in contrast to the original proposal. However, the experiments have demonstrated the possibility to control all system parameters with the required precision and stability. The addition of optical amplification can be considered an engineering problem for the future. The basic issues of the original idea have been addressed and solved with good control of the system in this proof of principle experiment. With respect to the presented results, I recommend to indicate in Figure 1, that the experiment did only have light optics, but no amplifier.

Per the referee's suggestion, we have modified the conceptual schematic in Figure 1 to show an additional set of non-amplified ("passive") optics with the label "this experiment." As this was a general conceptual diagram for the transit-time OSC method, we thought it important to leave the original configuration as well.

The authors describe their experimental methods and observations in a clear way, the figures prove the claim of the observation of cooling and are of good quality. The methods to prepare the system for cooling and the methods to detect the cooling are thoroughly described. The theoretical background for analysis of the measurements is described in other publications, but shortly summarized in the methods section. For a first experiment of optical stochastic cooling the agreement between theoretical calculations and experimental results is good, confirming the excellent mastering of the experimental equipment.

One aspect of their experiment might deserve a more detailed description. The authors report coupling of orthogonal subspaces of the 6-dimensional phase space. This technique is not considered in the original proposal and is only shortly mentioned in the manuscript. The method of coupling and the observation and control of coupling could be described in a chapter in the methods section in more detail.

This was described briefly in the text; however, per the referee's suggestion, and in the interest of shortening the main text, we have moved these comments to a dedicated paragraph in the Methods section: "Coupling of the phase-space planes."

In equation (14) the formula describes the ratio of damping times, although the text states rates. There is no definition of the damping time in the manuscript. This part should be modified to avoid the use of damping time or the damping time should be defined.

We have eliminated the use of damping time in favor of the rates. Furthermore, on inspection, the use of " $R_{v\tau}$ " from Eq. (14) as it conflicts with the previous use in Eq. (8), where it is the ratio of the small-amplitude OSC rate to the SR cooling rate. We have adjusted the text and equations to eliminate this conflict.

I recommend to consider the proposed modifications. The manuscript is already of good quality and the modifications are only intended to clarify details.

In general, the manuscript describes excellent experimental results which can be of big importance for strategies to improve accelerator systems, particularly high energy colliders and low emittance, high brilliance light sources. The results have relevance both for accelerator designers and users of the accelerators. The description is appropriate for a broad readership.

In conclusion, I strongly recommend the manuscript for publication in Nature.

We greatly appreciate the referee's supportive remarks.

Some minor comments and corrections:

line 42-44: for colliders, I would mention the increase of luminosity by shrinking the emittance in the course of cooling We have modified this sentence as follows: "In the case of colliders, cooling increases luminosity through the reduction of beam emittances and is essential for combatting intrabeam scattering (IBS) and other diffusion mechanisms^{11,12} "

line 400: 'the bunch length barely fit' -> 'the bunch lengths barely fit' or 'the bunch length barely fits'
This was changed to 'the bunch lengths barely fit'.

equation (12): epsilon_z should be epsilon_s

This correction has been made.

In the reference list I found misspelling of author names:

ref. 22, 43, 44: Zimmermann

ref. 29 Brennan

These misspellings have been corrected.

Other changes by the authors:

While updating the figures, we found a minor sign error in the worksheet that generated Extended Data Figure 3. This expression was used in one estimate of the equilibrium longitudinal amplitude based on the measured OSC strength. The error had minimal impact on the equilibrium amplitude, ~3% ($a_s = 3.273$ vs $a_s = 3.382$), and it does not modify any conclusions of the paper. Extended Data Figure 3 and the listed value of a_s in the methods section have been updated accordingly.

The intensity maps of Figure 3a were rescaled to ensure that the full range of the colormap was used. Reference 39 was corrected to meet the required style guidelines.

We are grateful to both referees for their time, effort and thoughtful responses.

Best regards,

J. Jarvis, V. Lebedev, A. Romanov, D. Broemmelsiek, K. Carlson, S. Chattopadhyay, A. Dick, D. Edstrom, I. Lobach, S. Nagaitsev, H. Piekarz, P. Piot, J. Ruan, J. Santucci, G. Stancari, A. Valishev